# A Novel Modeling Optimization Approach for a Seven-Channel Titania Ceramic Membrane in an Oily Wastewater Filtration System Based on Experimentation, Full Factorial Design, and Machine Learning

**DOI:** 10.3390/membranes14090199

**Published:** 2024-09-20

**Authors:** Mohamed Echakouri, Amr Henni, Amgad Salama

**Affiliations:** Process Systems Engineering, Produced Water Treatment Laboratory, Faculty of Engineering and Applied Science, University of Regina, Regina, SK S4S 0A2, Canada; echakoum@uregina.ca (M.E.); amgad.salama@uregina.ca (A.S.)

**Keywords:** oily wastewater, titania ceramic membrane, fouling control, optimization, statistical modeling, full factorial design, Multiple Linear Regression, Artificial Neural Network

## Abstract

This comprehensive study looks at how operational conditions affect the performance of a novel seven-channel titania ceramic ultrafiltration membrane for the treatment of produced water. A full factorial design experiment (2^3^) was conducted to study the effect of the cross-flow operating factors on the membrane permeate flux decline and the overall permeate volume. Eleven experimental runs were performed for three important process operating variables: transmembrane pressure (TMP), crossflow velocity (CFV), and filtration time (FT). Steady final membrane fluxes and permeate volumes were recorded for each experimental run. Under the optimized conditions (1.5 bar, 1 m/s, and 2 h), the membrane performance index demonstrated an oil rejection rate of 99%, a flux of 297 L/m^2^·h (LMH), a 38% overall initial flux decline, and a total permeate volume of 8.14 L. The regression models used for the steady-state membrane permeate flux decline and overall permeate volume led to the highest goodness of fit to the experimental data with a correlation coefficient of 0.999. A Multiple Linear Regression method and an Artificial Neural Network approach were also employed to model the experimental membrane permeate flux decline and analyze the impact of the operating conditions on membrane performance. The predictions of the Gaussian regression and the Levenberg–Marquardt backpropagation method were validated with a determination coefficient of 99% and a Mean Square Error of 0.07.

## 1. Introduction

Produced water (PW) is oily wastewater generated by various industrial activities, most notably within the oil and gas sectors. These industries not only produce significant quantities of crude oil but also generate vast volumes of wastewater. PW is a complex mixture comprising water, dissolved and undissolved oils, aromatic compounds, hydrocarbons, gases, organic materials, salts, heavy metals, suspended solids, bacteria, corrosion inhibitors, treatment chemicals, residual production-line additives, toxic pollutants, and other chemical substances [1,2,3,4]. Due to the environmental hazards posed by PW, stringent environmental regulations have been implemented, necessitating the treatment of PW to meet established discharge limits [5]. The search for novel, inexpensive PW treatment technologies that are simple to use and easy to maintain has recently gained prominence. Membrane technology has grown in popularity as a viable filtration approach for oily wastewater treatment [6]. Membrane filtration is commonly associated with low energy consumption, cost-effectiveness, and minimal footprint characteristics [7]. Regardless of its operating potential, membrane technology’s growing adoption is hampered by membrane fouling. Fouling is the deposition of rejected oil, pollutants, colloidal particles, and bacteria over the membrane surface, causing surface deterioration, a decline in the membrane permeation rate, and a rise in filtration costs [8]. The significance of fouling in the membrane technology field is well-known and has been frequently investigated. Researchers typically built physical, empirical, or semi-empirical models to analyze the process mechanism under various conditions. However, these models often fail to explain and monitor the data variation patterns under the operating conditions for a complex system with a complex feed composition [9], ceramic membrane pore tortuosity-blocking process [10], and turbulent hydrodynamic conditions. Therefore, understanding the membrane filtration system characteristics, properties, and components is insufficient for understanding membrane fouling. Various mechanistic models have been proposed, from dead-end to crossflow filtration, to describe the fouling phenomenon, but these are limited by the concentration polarization pre-fouling development [11]. As a result, it is essential to develop alternative methodologies to overcome the complexity of the filtration system, such as statistical modeling and Artificial Intelligence methods, to assist fouling system modeling by avoiding the intricate theories that underpin the process. These soft-computing methods are reliable, with problem-solving abilities and cost-effectiveness, and are result-oriented [12].

Statistical modeling approaches [13,14,15,16,17,18,19,20,21] are widely used to estimate the correlation between explanatory variables and response values in polynomial regression equations. Studying different operating conditions one by one is time-consuming and costly [22]. A statistical method such as full factorial design can circumvent this by looking at all the parameters and their impact for each level of operating condition [23,24]. Outcomes at all combinations of factor levels are measured. Variation of factor levels is performed, instead of one at a time, to explore all interactions and their effect on target responses. In this study, each experimental run factor was performed at two levels. All sequences of the factor levels are integrated into the experimental runs. The 2-level factorial design thoroughly investigates the factors domain, giving important information for a few runs per factor [25].

Artificial intelligence (AI) techniques, such as Multiple Linear Regression (MLR) and Artificial Neural Networks (ANNs), can be a viable choice for predicting the process design response based on the operating input parameters [26,27,28,29,30,31,32,33,34,35,36,37,38]. The MLR can map a nonlinear process and estimate the response values for multiple regressors using a polynomial equation [39,40]. ANN has been shown to be an eminent technique because it iterates, learns, and stores data to forecast the system’s response effectively. A fundamental advantage of AI modeling is its independence of the filtration process and capability to predict the system output without any prior knowledge or information about the feed characteristics or the membrane surface properties [40]. ANN models are mainly trained using the operating conditions and collected experimental data. Due to ANN’s flexible application and fitness to learn from the data, it has gained a great deal of acceptance in the wastewater management field. The ANN can be trained with the Levenberg–Marquardt (LM) Backpropagation algorithm (trainlm) [39]. A two-layer feed-forward network with hidden sigmoid neurons and one linear output neuron (fitnet) trains the network to fit the inputs and targeted response. Given reliable data and enough neurons in its hidden layer, the training algorithm can arbitrarily fit multi-dimensional variants mapping problems [41]. Moreover, it automatically stops when generalization stops improving. ML-ANN backpropagation successfully delivers a practical regression approach and analysis of the operating conditions’ variations for predicting the permeate flux in oily wastewater systems compared to other standard modeling techniques.

In this study, the assessment of the full factorial design, Multiple Linear Regression (MLR) analysis, and neural network techniques for the predictions and optimization of the operating conditions (TMP, CFV, and FT) of an ultrafiltration (UF) titania ceramic membrane was performed, treating a synthesized produced water feed representing the PW of the Bakken reservoir in Western Canada. An analysis of variance was performed to determine the significant effects of both univariate and multivariate operating conditions on the membrane permeate flux and yield permeate volume. The response values and their sensitivity variation to the operating conditions were visualized using contour plots, highlighting the optimal process operating regions. The composite desirability function was used as an optimization method to assess the operating variables settings and maximize the permeate flux and volume [4]. Additionally, the membrane’s rejection capacity was analyzed and reported for each experimental run under various operating conditions. Finally, MRL- and ANN-trained models were used to fit the membrane permeate flux decline data and provide an algorithm for forecasting the system behavior under various operating parameters. The models’ statistical performance was validated using the correlation coefficient and Root Mean Square Error as the statistical key metrics.

Table 1 gives an overview of published filtration studies for oily wastewater found in the literature with details of the operating variables examined, the optimization techniques employed, and the resulting performance outcomes.

## 2. Theory

For the analysis of the experimental data, a full factorial design [42,43,44,45], multilayer regression model algorithms (MLRs), and artificial neural network (ANN) [46,47,48] were selected to predict the membrane permeate flux. Recently, researchers have extensively applied artificial intelligence to optimize the operating conditions of the process, targeting cost-effectiveness and time-saving experimental design [49,50,51,52,53,54].

### 2.1. Full Factorial Design Methodology

This study employed a 2^3^-full factorial design to screen the impact of the transmembrane pressure, crossflow, and filtration time, as the primary independent factors, on membrane performance. Additional factors such as pH, feed oil content, and temperature were maintained constant during this experiment [1]. The studied factors are often evaluated in the literature and considered the most critical parameters in membrane process optimization [2,13,14,15]. The levels and symbols of each independent factor were set for high (+1), middle (0), and low (−1), as reported in Table 2. The factors’ levels were combined to capture the full interactions of the operating conditions and their impact on the membrane performance. A total of 11 runs were performed to obtain a quadratic regression model of 8 runs and 3 center points (Equation (1)). The experiments were performed randomly to minimize experimental errors [55]. The factor ranges were identified based on a review of the literature [2,4,14,15,55], the setup capabilities, and considering experimental costs. Table 2 illustrates the coded levels of each independent factor for the full factorial design experiment.

A robust data analysis, statistical, and process improvement software tool, Minitab 19, was used to predict, visualize, and analyze the experimental data. A quadratic regression model was developed to predict the response values [4]:(1)$$Y_{k\;}=b_{0}+\sum_{i=1}^{n}b_{i}\;x_{i\;}+\sum_{i<j}^{n}b_{ij}\;x_{i}\;x_{j\;}+\sum_{i=1}^{n}b_{ii}\;x_{i}^{2}+S_{e}$$
where *Y_k_* (*k* = 1, 2) is the predicted response used as a dependent variable to predict *Y*_1_: the permeate flux and *Y*_2_: permeate volume, respectively. *x_i_* (*i* = 1, 2, and 3) designates the selected factors, *x*_1_: TMP, *x*_2_: CFV, and *x*_3_: FT, respectively. In this equation, *b*_0_ is a model constant coefficient, and *S_e_* is the statistical error (residual); *b_i_* (*i* = 1, 2, 3), *b_ij_* (*i* = 1, 2, 3; *j* = 1, 2, 3; *i* < *j*), and *b_ii_*, denote the linear, interactional, and quadratic regression model coefficients, respectively. These coefficients are approximated by Multiple Linear Regression analysis. The goodness of fit of the full factorial polynomial model was quantified with the coefficient of determination (R^2^) and the lack of fit. A variance analysis was carried out to ascertain the significance of the regression factors and their interactions. A significant level *p*-value for the model below 0.05 was deemed meaningful and considered significant at more than 95%. The measured responses for the permeate flux decline and permeate volume were denoted J_ni_ (final steady permeate flux measured in LMH) and Y_nf_ (final net permeate volume measured in litres (L)). In each experiment, the permeate flux decline, steady-state total permeate volume (Table 3), rejection capacity, and permeate turbidity (Table 4) were used to quantify the membrane performance and fouling control.

### 2.2. Multiple Linear Regression Method

The Multiple Linear Regression (MLR) model is commonly employed as an empirical model to estimate and explore the functional link between the dependent responses’ target values and the independent explanatory regressor variables. MLR is employed when more than one predicted response variable is involved. MLR analysis prompts estimating the regression coefficients of the fitted linear model equation by the ordinary least squares method to predict the experimental data. A Multiple Linear Regression model (Equation (2)), with n independent variables, *x_i_* (*i* = 1, …, *n*), and *y_i_* (*i* > 1) dependent response, is written in terms of the regression coefficients as follows [39,56]:(2)$$y_{i\;}=\mu_{0\;}+\sum_{j=1}^{n}\mu_{j}x_{ij}+e_{i},\;i=1,\;2,\;\ldots ,\;m$$
where *µ*_0_ and *µ_j_* (*j* = 1, …, *n*) are the regression coefficients and *e_i_* is the model regression error.

The least squares method finds the best estimation coefficient *µ_j_* in Equation (2) for the descriptive data set to minimize the sum of the squared errors. The least squares function is represented by Equation (3) or Equation (4):(3)$$S=\sum_{i=1}^{m}e_{i}^{2}$$
(4)$${S=\sum_{i=1}^{m}\left(y_{i}-\mu_{0\;}-\sum_{j=1}^{n}\mu_{j}x_{ij}\right)}^{2}$$

The objective least squares function *S* is minimized according to the regression coefficients *µ*_0_, *µ*_1_, *…*, *µ_n_.* The terms’ coefficients *b*_0_, *b*_1_, *…*, *b_n_*, fulfill the Equations (5) and (6), respectively [56].
(5)$$\frac{\partial S}{\partial \mu_{0}}=-2\sum_{i=1}^{m}\left(y_{i}-b_{0}-\sum_{j=1}^{n}b_{j}x_{ij}\right)=0$$
(6)$$\frac{\partial S}{\partial \mu_{j}}=-2\sum_{i=1}^{m}\left(y_{i}-b_{0}-\sum_{j=1}^{n}b_{j}x_{ij}\right)\;x_{ij}=0$$

Data for the eleven experimental runs were used to train the regression models, including the linear regression models, regression trees, Support vector machines, Gaussian process regression models, and Ensembles of regression trees (Appendix A). The regression models’ data inputs were the TMP, CFV, and FT of each experimental run, and the response was the permeate flux. Matlab R2020b was used as a technical computing tool for data analysis and visualization. Nineteen regression models were trained (Appendix A) by applying cross-validation folds of five to avoid the model overfitting. Then, the performance was compared side-by-side using the statistical parameters R-squared, RMSE, MSE, and MAE (Equations (7)–(12))
R-squared (the coefficient of determination) is a goodness-of-fit measure for strength linear regression association between the trained model response and experimental data. The closer to one, the better the model is.
(7)$$R^{2}=1-\frac{RSS}{TSS}$$
(8)$$RSS=\sum_{i=1}^{n}{\left({Flux}_{i}-\hat{Flux}\right)}^{2}$$
(9)$$TSS=\sum_{i=1}^{n}{\left({Flux}_{i}-\bar{Flux}\right)}^{2}$$where *RSS* = Residuals sum of squares; *TSS* = Total sum of squares; $\bar{Flux}$ = mean value of the experimental data set; $\hat{Flux}$ = predicted flux value; and *Flux_i_* = the true flux value.

*RMSE* (Root Mean Square Error) is the root mean of residuals (model error prediction). *RMSE* depicts the standard deviation of the residuals around the model regression fit. The smaller the value, the better the model is (Equation (10)).


(10)
$$RMSE=\sqrt{\frac{\sum_{i=1}^{n}{\left({Flux}_{i}-\hat{Flux}\right)}^{2}}{n}}$$


*MSE* (Mean Squared Error) is the average squared residuals between the predicted and observed paired response values. *MSE* is the square of the *RMSE*. The smaller the *MSE*, the better the regression model (Equation (11)).


(11)
$$MSE=\frac{1}{n}\times \sum_{i=1}^{n}{\left({Flux}_{i}-\hat{Flux}\right)}^{2}$$


*MAE* (Mean absolute error) is an average of the positive residuals between predicted and actual responses. The smaller the *MAE*, the better the regression model (Equation (12)).


(12)
$$MAE=\frac{1}{n}\times \sum_{i=1}^{n}\left|{Flux}_{i}-\hat{Flux}\right|$$


### 2.3. Artificial Intelligence Method

An important use of the ANN technique is the prediction of the process’s best operating parameters when one or more input variables vary [12,39,53,57,58]. In this study, ANN models were built by varying three input parameters: TMP, CFV, and FT [12,51,59] to predict the membrane permeate flux.

ANNs are artificial intelligence computational algorithms inspired by human biological neuron systems. ANNs are extensively used in experimental design to reduce costs and optimize conventional process performance [60]. ANNs are applied in numerous engineering fields to solve supervised and unsupervised learning problems such as classification, regression, and clustering [61,62]. It is vital to investigate the architecture of the artificial neural network to better comprehend its operation (Figure 1) [39]. ANN architecture is a neural topology connecting the input neurons and the output. These neurons are linked together and layered to construct a nonlinear operating system. The amount of input data defines the number of the neurons’ input layer. The topology of the ANN network mostly depends on the number of hidden layers, neurons in each layer, and the type of transfer function processing the input data. The final output layer performs the final output computation [53]. This study selected to use of the feed-forward Levenberg–Marquardt (LM) backpropagation training algorithm [63,64,65]. This transfer function uses a Jacobian for simulations, assuming a control performance function (MSE) [66]. The algorithm looks to be the quickest way to train moderate-sized backpropagation neural networks, up to several hundred weights. It is also easy to implement in MATLAB software since the matrix equation solution is a built-in function. As a result, its characteristics become much more evident in a MATLAB setting [61].

The ANN training stops when generalization halts learning, as quantified by an increase in the Mean Square Error and a decline in the model performance [62]. In supervised training, for a given input/output data set, the ANN algorithm can provide an optimal mapping process to predict the collected targeted experimental data based on the independent explanatory operating conditions parameters [66]. The iterative neural network learning process builds a probability-weighted relationship between the input and output data. Data are stored within the neural network throughout the training while comparing the experimental target response and model predicted value. The ANN adjusts the weight coefficients to minimize the error intricated by the selected transfer function until the threshold is respected. Successive adjustments will be performed until the training algorithm fits the best target output. The Matlab ANN Toolbox was mainly employed in this study analysis. The ANN architecture used in this study is presented in Figure 1.

To apply the ANN to the collected data, we investigated the influence of variation in the number of neurons on the training performance. A trial-and-error method was applied to determine the optimum number of neurons. Numerous models with incremental hidden layer numbers of neurons from 5 to 60 were studied, and their Mean Squared Error (MSE) was documented and compared. Figure 2 identified the minimum MSE percentage for twenty neurons as the best ANN network configuration. Another performance parameter is the R-squared (coefficient of determination (R^2^)), which designates the model’s goodness of fit to predict the collected experimental data. MSE and R-squared are parameters used to select the best model that fits the experimental data set.

Consequently, an ANN model with 20 neurons was employed for further simulation and analysis. A total number of experimental data for samples between 1200 and 2400, depending on the filtration time variable (one to two hours), were considered for each model analysis. In this study, the gathered experimental data were divided into 80% for the training model, 10% for validation, and 10% for testing.

For model performance comparisons, the Mean Squared Error percentage (*MSE* (%)) is computed using the following Equation (13):(13)$$MSE\;(\%)=\frac{MSE}{Max\;(flux)}\times 100$$

## 3. Materials and Methods

### 3.1. Materials

A seven-channel ceramic membrane of 25 mm diameter was acquired from Tami Industries (Nyons, France) and cut into 305 mm pieces to match the membrane housing unit size. Bakken light oil from western Canada (Saskatchewan) was used in all experiments with a measured viscosity of 5.23 cP and a density of 0.87844 g/cc at 22.5 °C. Reverse osmosis (RO) was utilized to prepare the ultrapure deionized water (DI, <5 ppb TOC and <0.1 colony-forming units of a microorganism/mL) using an ultraviolet (UV) water purification system (EMD Millipore, 2012, Burlington, MA, USA). The supplementary chemicals for the ultrafiltration ceramic membrane regeneration and Horiba S-316 oil/solvent extraction were purchased and used as received (Table 5).

### 3.2. Feed Synthesis and Characterization

The laboratory-synthesized produced water feed of ~200 ppm was prepared by mixing reverse osmosis water, Bakken light oil, and Sodium dodecyl sulfate (SDS), used for oil droplet stability, in the ratios of 2 L, 4.5 mL, and 0.3 mM, respectively. The 2 L of PW emulsion was mixed for two minutes using a commercial blender at 19,000 rpm with variable pulses to achieve a homogenous and stable feed mixture. This procedure was repeated for a total volume of 24 L (12 batches), as required for each experiment. The feed water properties, such as oil content, chemical oxygen demand (COD), turbidity (NTU), pH, zeta potential, feed mean droplet size, density, and viscosity, were measured and are reported in Table 6. The experimental instruments employed for feed water characterization are listed in Table 7.

### 3.3. Ceramic Membrane Characterization

The seven channels of a UF ceramic membrane (Appendix A) were used to treat synthesized water feed using the LabBrain filtration unit fabricated by LiqTech International (Hobro, Denmark). Fundamental membrane characteristics and properties such as ceramic membrane morphology, pore size, molecular weight cutoff (MWCO), retention capability, permeability, ceramic membrane geometry/dimensions, and thermal/chemical resistance are reported in Table 8. The ceramic membrane surface wettability is investigated by measuring the contact angle of water and oil droplets’ adhesion to the membrane surface. Appendix A depict the membrane surface super-hydrophilicity and oleophobicity properties with a contact angle of 35 and 135°, respectively.

### 3.4. Crossflow Filtration System Description

In this study, a LabBrain CFU022 automated crossflow ceramic membrane filtration unit was employed to perform eleven batch-mode experiments (Appendix A). Appendix A presents a schematic diagram of the LabBrain Proportional and Integral (P&I) control loop diagram system. The unit can operate in three modes, either manually or automatically: constant transmembrane pressure, constant feed crossflow, and constant permeate flow. To enhance the ceramic membrane’s high permeability water flow, the UF membrane was pre-experimentally prepared by a progressive submersion in deionized water and then left in a bath for an additional twelve hours to fully evacuate any trapped air from the membrane pores. The membrane element was installed and fastened to the cylindrical housing cell. A 24 L feed water emulsion was synthesized, characterized, and immediately added to the oily wastewater container to run the experiment. The speed pump level and the concentrate valve opening percentage were used to gradually adjust the settings of the filtration unit for CFV and TMP operating values. All data and operating parameters of the valves’ actual openings, transmitters, transmembrane pressure, and flux were automatically logged into the unit’s internal memory every 3 s. The concentrate was directed to the feed tank for continuous filtration. Once measured, the permeation flow is returned to the oily water container to keep a constant feed concentration. The membrane oil rejection performance was calculated using the following Equation (14):(14)$$Rejection\;\left(\%\right)=\left(1-\frac{C_{p}}{C_{f}}\;\right)\times 100$$
where *C_f_* and *C_P_* are the oil content of the feed and permeate, respectively, in ppm.

Following each experiment (Appendix B), the filtration system was carefully rinsed and drained using reverse osmosis water (50 °C), Sodium hydroxide (50 °C), and Phosphoric acid (85 °C), respectively, to eliminate oil, pollutants, and impurities until reaching the ceramic membrane initial flux recovery of 99% (Appendix A).

## 4. Results and Discussion

Membrane filtration in crossflow mode was often investigated by researchers as a prominent technique for fouling control and mitigation [67]. This study is a comprehensive investigation of the impact of operating conditions on the membrane performance. The optimal conditions were examined using full factorial design, Multiple Linear Regression analysis, and Artificial Neural Networks. A regression model is presented for the permeate flux and volume for the full factorial design. The MLR was employed for learning regression analysis; several algorithms were tested to select the best-fitting model and predict the permeate flux. Finally, the Artificial Neural Network-based approach was applied to fit the membrane flux decline data.

### 4.1. Full Factorial Design Analysis

#### 4.1.1. Membrane Performance

This study investigated three operating parameters to examine their influence on two response parameters, i.e., the membrane flux and the steady-state permeate volume. Eleven experimental runs were performed, and the membrane rejection and permeate turbidity were recorded. The titania ceramic membrane demonstrates an excellent rejection capability of between 91% and 99% for the different operating sets of conditions (Table 7 and Table 8). This performance is similar to the reported research outcome in [68,69] for similar ceramic membranes.

Each experiment rejection test was performed to evaluate oil rejections, amongst others (Table 8). Experimental runs 5 (+1, +1, +1) and 6 (+1, +1, −1) demonstrate the best oil rejections, attaining 97% and 98% for an overall permeate oil content <= 3 ppm, respectively. Similarly, these runs also match those with the highest permeate flux and permeate volume of 297 and 341 LMH and 8.14 and 7.82 L, respectively.

For an initial oil feed of ~200 ppm, the permeate oil content was below ~5 ppm for all the experimental runs. A configuration of a TMP of 1.5 bar (level +1) and a CFV of 1 m/s (level +1) for a filtration time of either one or two hours (level −1 or +1) led to the best performance for the titania membrane. The optimized design was achieved for (+1, +1, +1), with an overall membrane performance, flux, and permeate volume of 297 LMH and 8.14 L, respectively. Measurements were performed twice, and the averages with the standard deviations were reported.

#### 4.1.2. Statistical Analysis

To understand the effect of the operating conditions (TMP, CFV, and FT) and their interactions on the membrane performance and fouling mitigation. An Analysis of Variance (ANOVA) was performed to illustrate the results of the full factorial design experiment. Table 9 and Table 10 summarize the analysis of variance for the membrane flux and permeate volume, respectively. The results of *p*-values for each model depicted that the values are closer to zero, reflecting a higher significance of the regression models for both responses. For a CI of 95%, a *p*-value below 0.05 means, statistically, that the explanatory factor or their interaction is significant. The *F*-value demonstrates the degree of the impact of these factors on the membrane’s overall performance. For a CI of 95%, the empirical regression models for the two responses are presented by the following polynomial Equations (15) and (16):(15)$$\begin{array}{cc} Flux\;(LMH) & =210.75+56.75\;TMP+35.75\;CFV-11.5\;FT\\ & +15.75\;TMP\times CFV-2.00\;TMP\times FT-3.00\;CFV\times FT\\ & -5.50\;TMP\times CFV\times FT \end{array}$$
(16)$$\begin{array}{cc} Permeate\;(L) & =7.031+0.606\;TMP+0.184\;CFV+0.149\;FT\\ & +0.159\;TMP\times CFV-0.026\;TMP\times FT-0.019\;CFV\times FT\\ & +0.056\;TMP\times CFV\times FT \end{array}$$

The coefficients of determination (R^2^) of the two models, flux and permeate volume, were 99.99% and 99.98%, respectively. The adjusted terms (R-sq(adj)) were 99.95% and 99.88%, respectively, indicating that the two proposed models predicted well more than 99% of the experimentally collected data [70].

Figure 3 presents the Pareto chart to demonstrate the absolute values of the standardized effects of the terms T-values in descending magnitude effect. The reference line indicates the factors’ significant statistical effects on the responses’ values. The critical T-values for the membrane flux and permeate volume were both 4.30 at a CI of 95%. In the Pareto chart, the explanatory factors and their interactions that cross the reference line are statistically significant. The regressor parameters’ effect and their interactions are readily exhibited in Figure 3, where A > B > AB > C > ABC > BC for flux and A > B > AB > C > ABC for the permeate volume.

The factorial plots for flux and permeate volume (Figure 4) provided a complete visualization of the two-way interaction effects between the independent predictor’s factors and the mean response values. In each interaction block, the main effect of the regressors exists when the lines (solid and dashed) are not horizontal, indicating that the mean response values vary with the predictors’ interactions. The steeper the lines’ slope, the higher the main effect magnitude and the interaction between the explanatory variables.

Figure 4a shows, in the top left, that the relationship between the membrane flux and TMP depends on the CFV variation. For CFV at the higher level +1 (the dashed green line), the mean flux varies from 174 to 319 LMH (145 LMH increase). In contrast, at lower level −1 (the solid blue line), the membrane flux varies from 134 to 216 LMH (82 LMH increase) with the variation of TMP from the −1 to the +1 level.

For the TMP at the lower level, −1, the variation of the CFV from the lower level, −1, to the higher level, +1, increases the mean flux from 134 to 174 LMH (40 LMH variation). Whereas, at the TMP with a higher level of +1, the variation of CFV from a lower to a higher level increased the mean flux from 216 to 319 LM (103 LMH variation). This indicated that the interaction between the CFV and TMP tremendously affects the membrane flux. Similarly, in Figure 4b, in the top left, the interactions of the TMP × CFV were significant. However, the interaction of TMP × FT and CFV × FT was insignificant (Figure 3b). Thus, the three-way interactions of TMP × CFV × FT were crucial and significant at CI 95% for both responses: membrane flux and permeate volume (Table 9 and Table 10 and Figure 3 and Figure 4).

To provide statistical evidence of the model goodness of fit to predict the experimental data, a normal distribution of the collected data is assumed [71]. The mentioned assumption can be verified using one of the following statistical tests: Anderson–Darling Test, Ryan–Joiner Test, Shapiro–Wilk Test, Kolmogorov–Smirnov Test, Scatter Plot, or Normal Probability Plot. In this study, the normal probability plot (P-P) for membrane flux and permeate volume was performed at a CI of 95%, as depicted in Figure 5, respectively. A comparison between the *P*-value to the significance level (0.05) in the P-P plot indicates that the null hypothesis to reject the normality of the data fails (*p*-value > 0.05). There is insufficient evidence to conclude that the experimental data are not normally distributed. In addition, a visualization of the P-P plot reasonably indicates that all the data points are linearly close to the linear regression line and within the confidence interval of 95%. Therefore, the experimental data are normally distributed at a CI of 95%.

Figure 6a and Figure 7a display the impact of the CFV and TMP on the membrane flux and permeate volume, respectively. From the plots contour, the flux and permeate volume are enhanced by increasing the TMP and CFV simultaneously.

Figure 6a illustrates that increasing the crossflow velocity while maintaining a constant TMP above 1.25 bar positively affects permeate flux. A higher crossflow velocity creates turbulent conditions at the membrane surface, which reduces the concentration polarization layer and helps mitigate fouling by sweeping away accumulated oil droplets. This improved turbulence enhances permeate flux, allowing the membrane to operate more efficiently. As a result, more of the feed solution passes through the membrane, leading to a greater overall permeate volume and increased productivity while sustaining filtration performance over time. However, if the operating conditions are below those of the membrane’s critical flux, raising the crossflow velocity at this limited flux will have a diminishing effect on permeate flux and volume. Since the flux is already below the critical threshold, the driving force for permeation is restricted. In such scenarios, although higher crossflow velocity can still reduce fouling and improve membrane cleaning, the overall increase in permeate flux and volume will be minimal due to the system already operating under conditions that prevent significant flux improvement. Essentially, without adequate flux to drive the filtration process, the advantages of increased crossflow velocity are limited (Figure 6a). In Figure 6a, at a constant pressure of 0.9 bar, increasing CFV from 0.65 m/s typically improves the flux from 250 to 300 LMH, as it helps to maintain a cleaner membrane surface and reduces concentration polarization. However, high CFV beyond 0.85 m/s with relatively low transmembrane pressure may cause turbulence and lead to higher energy losses without a proportional increase in flux. This may explain the effective flux decline due to energy inefficiencies or hydraulic limitations.

The operation at the highest TMP of 1.5 bar (level +1) ensures a higher permeability due to the pressure-driven dragging permeation force of the feed toward the membrane surface. However, this is also associated with rapid oil droplet accumulation at/within the membrane pores, enhancing a concentration polarization barrier toward the continuous filtration process [72]. Coupling the TMP and CFV impacts the permeate flux, which appears to be enhanced with increasing CFV (to 1 m/s) (Figure 6a). Crossflow velocity creates turbulent hydrodynamic conditions at the membrane surface to boost cleaning by sweeping the oil droplets in the main crossflow field [73,74].

Figure 6b demonstrates that flux increases with higher CFV, particularly in the optimal range of from 0.8 to 1.0 m/s, due to the enhanced shear forces that reduce fouling. Over time, flux generally declines, as shown by the lighter green regions, especially at lower CFV (below 0.7 m/s), reflecting the typical impact of fouling on membrane performance [8,75]. The plot aligns with the understanding that maintaining a higher CFV helps sustain higher flux by mitigating fouling, confirming the importance of optimizing CFV for efficient filtration in constant TMP scenarios. A distinct region with high flux (dark green) is also observed when CFV is around 0.9 m/s and filtration time is between 1.2 and 1.6 h, followed by a gradual decline in flux over time, especially at lower CFV values, which suggests that prolonged filtration without sufficient CFV leads to a decrease in filtration efficiency.

A similar analysis can be performed for the contour plot of permeate versus TMP and CFV at constant FT (middle level = 1.5 h). Figure 7 illustrates that, when TMP increases, the permeate generally increases. At lower CFV levels (0.5 m/s), the increase in TMP from 1.0 to 1.5 bar has a limited impact on the permeate volume due to fouling development. However, with a higher CFV, the concentration polarization and fouling mitigation were enhanced, and permeability was improved. It is mandatory to consider an optimized CFV value to avoid reducing the permeate due to increased shear forces or other operational factors. The plot suggests that the optimal permeate volume is achieved with higher TMP and moderate CFV between 0.8 and 1.0 m/s, providing valuable insights for optimizing operational conditions.

The Interactions between Input parameters, TMP × FT for permeate flux, and TMP × FT and CFV × FT for permeate volume, showed no significant effect at the 95% confidence interval (CI). These results have been included in the Appendix A.

#### 4.1.3. Model Optimization Analysis

Model optimization settings were selected to increase the membrane’s final steady flux and permeate volume by determining the best operating conditions (TMP, CFV, and FT) (Table 11). TMP, CFV, and FT were set within their ranges. A multiple response prediction was performed to identify the two solutions (+1, +1, −1) and (+1, +1, +1) for maximum steady flux and permeate volume of 341 LMH and 7.82 L and 297 LMH and 8.14 L for a composite desirability function of 0.916 and 0.894, respectively (Table 12). The two selected configurations of the operating conditions were similar (TMP of 1.5 bar and CFV of 1m/s). The second solution is just a continuous process operation for an extra hour. For the process configuration (+1, +1, +1), the filtration time is 2 h, the permeate volume is 8.14 L, and the final steady flux loss is 297 LMH. Operating at (+1, +1, +1) (Table 12) is characterized by an increase in permeate volume of 3.92% and a drop in the steady permeate flux of 12.90% compared to (+1, +1, −1). Operating the system beyond one hour shows more oil droplets accumulation at the membrane surface and a decline in the final permeate flux. In contrast, continuous filtration improves the overall permeate volume.

The selection of the best operating conditions can be based on the filtration time. The optimization plot (Figure 8) was used to identify the optimal operating configurations for the explanatory regressors, assuming the defined parameter-accepted boundaries. The interactivity of the optimization plot eases the examination of the effect of the variables on the predicted responses. For this experimental design, the operating conditions (+1, +1, +1) were selected, prioritizing the permeate volume criterion, while the steady flux level was still high and acceptable compared to (+1, +1, −1).

As a result, Full Factorial Design provides a structured and systematic approach to experimentally explore the effects of multiple factors on the filtration unit. It involves testing all possible combinations of factor levels. Full Factorial Design helps in understanding how different factors and their interactions affect the filtration process by providing insights into main effects and potential synergies or dependencies for pinpointing the most effective filtration parameters. Full Factorial Design focuses on identifying optimal conditions for the filtration process based on the observed data. It grants insights into which factor levels contribute most to desired outcomes. However, Full Factorial does not involve learning from data in the way AI techniques, especially machine learning (ML), do.

Rather than conducting a one-time Full Factorial experiment, a sequential approach can be adopted where AI is used to guide subsequent experiments based on the observed results, creating a dynamic and adaptive optimization process. Both AI techniques and Full Factorial Design can be used together for a comprehensive analysis. AI/ML can learn and analyze available data generated from experiments, providing additional insights and optimization opportunities.

When comparing AI and Full Factorial Design for a filtration system, it is important to recognize that these approaches serve different purposes and can be complementary in most situations. The following will explore how the machine learning method can be applied in the context of a filtration system.

### 4.2. MLR Analysis

Firstly, the MLR analysis method was performed to predict each of the eleven experimental runs. The results from the MLR analysis of all experimental data are summarized in Appendix A. Nineteen training regression models were performed for each experiment, including linear regression models, regression trees, support vector machines, Gaussian process regression models, and ensembles of regression trees. For model training performance, the data were exported to the workspace, the features were selected, and validation schemes were specified fivefold. Then, the performances were compared side-by-side using their R-squared, RMSE, MSE, and MAE statistical parameters. Appendix A displayed that the best model that describes all the experimental data for the eleven runs is the Gaussian regression process (GRP) with the Rational Quadratic training function. The Gaussian model presents the greatest R-squared and smallest RMSE, MSE, and MAE values, respectively. The validated model (GRP) results were diagnosticated for the eleven experiments for further analysis. The statistical performance of the model was measured using R-squared, RMSE, MSE, and MAE to see how well the model fit the experimental data (Table 13). The results demonstrated a higher model’s goodness of fit to the data.

Experimental run #5 (+1, +1, +1) presented the best GRP model accuracy prediction performance. Additional diagnostic measures were performed to depict the model discrepancy between actual and expected values, such as the predicted versus actual plot (Figure 9). Furthermore, the response plots were analyzed to demonstrate the response fitted values as a function of experimental time (Figure 10a), TMP (Figure 10b), and CFV (Figure 10c), respectively. Figure 10 shows that the model accuracy is very high throughout the experimental time, aligning the actual and predicted results. The predicted permeate flux and the actual experimental data are error-zeroed at the steady stage filtration as a function of TMP or CFV, respectively. In contrast, the error at the early stage of filtration is very small due to the hydrodynamic environment of the filtration process (Figure 10b,c). The residuals as a function of TMP and CFV were also plotted. Figure 11a,b show that the residuals are close to zero value for all TMP and CFV during the two hours of experiments, respectively.

In conclusion, the GRP model can predict 99% of the permeate flux for all the experimental runs (R-squared is 99% for all the trained models by the GRP algorithm). The GRP model is beneficial for estimating the membrane permeate flux when one or more operating conditions of the filtration system vary (TMP, CFV, and FT) without conducting any experimental run (Table 13). A screenshot of the optimization models is shown in Appendix A.

### 4.3. ANN Analysis

This study has also used an ANN modeling approach to predict the membrane permeate flux for the eleven experiments. An ANN training algorithm with the Levenberg–Marquardt (trainlm) backpropagation method was used to train all the data based on a feed-forward network with one hidden and output layer. Before the ANN analysis, the number of neurons in the hidden layer was optimized to select twenty neurons as the best-designed network. Additional analysis was performed to check the model performance against the training data percentage. Data were trained for 60, 70, 80, and 90% of the size of the experimental values to investigate its impact on the model correlation. Figure 12 demonstrates that the data percentage and the model training performance are not correlated. Table 14 demonstrates that the sample size depends on the filtration time. Sample sizes were 1200, 1800, and 2400 for a filtration time of one hour (level −1), one hour and a half (level 0), and two hours (level +1), respectively. The sample size was randomly divided for all experimental runs into 80% for training, 10% for validation, and 10% for testing. For experimental runs #2, 3, 6, and 9, 80% (960 samples) were randomly chosen for training, 10% (120 samples) for validation, and 10% (120 samples) for testing. For runs #1, 4, and 8, 80% (1440 samples) were arbitrarily designated for training, 10% (180 samples) for validation, and 10% (180 samples) for testing. For runs #5, 7, 10, and 11, 80% (1920 samples) were similarly selected for training, 10% (240 samples) for validation, and 10% (240 samples) for testing. The eleven overall models’ performances were analyzed using the Mean Squared Error (MSE) and correlation coefficient (R-value) and the results are summarized in Table 14.

Appendix A displays the linear correlation between the experimental data and the model-predicted values for the training, validation, testing, and overall performance of the Artificial Neural Network (ANN) model across eleven experimental runs. Table 14 details the correlation coefficients and Mean Squared Errors for each experimental trial, providing quantitative insight into the model’s accuracy. The results demonstrate that the regression lines for all training models have slopes close to one and intercepts near zero, indicating a strong agreement between predicted and experimental data. Consequently, the ANN regression model achieved a goodness of fit of 99%, highlighting its high predictive accuracy for the experimentally observed membrane permeate flux under varying operating conditions. Overall, the ANN-based experimental design approach proves to be an effective and cost-efficient method for optimizing and enhancing the performance of membrane filtration systems in oily wastewater treatment applications.

## 5. Conclusions

In the present study, synthetically produced water based on brine from the Bakken reservoir in southern Saskatchewan, Canada, was used. The water contains an average feed oil content of approximately 200 ppm and was treated using an ultrafiltration titania ceramic membrane with a molecular weight cutoff of 150 kDa. The ultrafiltration membrane exhibited an excellent oil droplet rejection rate, ranging from 91% to 99%, across all operating conditions in the eleven experiments conducted. Initially, a full factorial design experiment was performed to assess the influence of operating conditions (TMP, CFV, and FT) on the membrane’s permeate flux decline and overall permeate volume reduction.

The optimal membrane performance was observed at a TMP of 1.5 bar (level +1) and a CFV of 1m/s (level +1) during a filtration time of from one to two hours (level −1 or +1). Under these conditions, the filtration system was optimized at (+1, +1, +1) to achieve a higher permeate volume of 8.14 L and a steady permeate flux of 297 LMH, corresponding to a flux decline of less than 38% from the initial value. This was performed in order to evaluate the effect of operational variables (TMP, CFV, and FT) and their interactions on membrane filtration efficiency and fouling mitigation. Analysis of Variance (ANOVA) was employed to analyze the outcomes of the full factorial design experiment. For a 95% confidence interval, two empirical regression models, flux and permeate volume, were developed, demonstrating excellent goodness of fit at 99.99% and 99.98%, respectively, for predicting the response values. The analysis of primary operating conditions and their interaction effects indicated as follows: (a) for membrane flux: TMP > CFV > (TMP × CFV) > FT > (TMP × CFV × FT) > (CFV × FT), and for permeate volume (b): TMP > CFV > (TMP× CFV) > FT > (TMP × CFV × FT).

Two predictive modeling approaches were utilized: Multiple Linear Regression (MLR) and Artificial Neural Networks (ANNs). In the MLR framework, nineteen distinct training regression algorithms were tested across eleven experimental runs. These algorithms included Linear Regression, regression trees, Support Vector Machines, Gaussian Process Regression, and Ensembles of regression trees. The models’ performance was evaluated using statistical metrics such as R-squared (R²), Root Mean Square Error (RMSE), Mean Squared Error (MSE), and Mean Absolute Error (MAE) to identify the best algorithm for predicting the experimental data. Among the models tested, the Gaussian Process Regression (GPR) utilizing the Rational Quadratic training function emerged as the optimal MLR model for estimating permeate flux across all experimental conditions. It is noteworthy that the GPR model predictions demonstrated a strong agreement with the experimentally observed data, indicating a satisfactory level of predictive accuracy.

An ANN training algorithm, utilizing the Levenberg–Marquardt backpropagation method, was employed to train all eleven sets of experimental data under varying operating conditions. The network architecture comprised a feed-forward network with a single hidden layer of 20 neurons and one output layer. The ANN-trained model was found to be a cost-effective approach to optimizing and improving the UF membrane performance by predicting the steady-state permeate flux decline during the experiment. The optimized model achieved a coefficient of determination (R²) of 99% and a Mean Squared Error (MSE) of 0.067, indicating its high predictive accuracy.

## Figures and Tables

**Figure 1 membranes-14-00199-f001:**
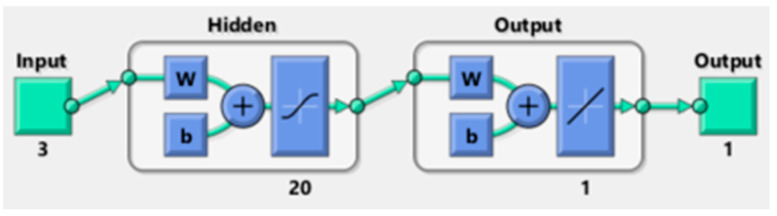
Function fitting neural network view.

**Figure 2 membranes-14-00199-f002:**
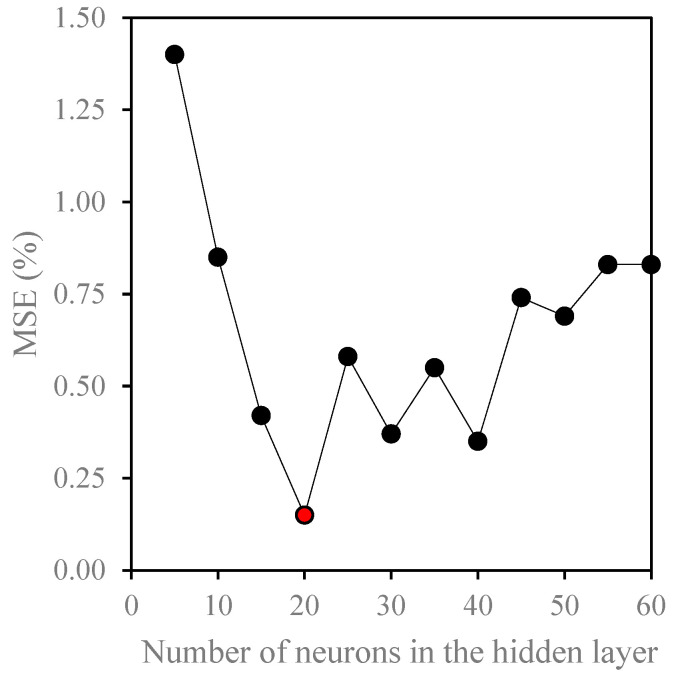
Effect of the number of neurons in the ANN hidden layer on the model training performance.

**Figure 3 membranes-14-00199-f003:**
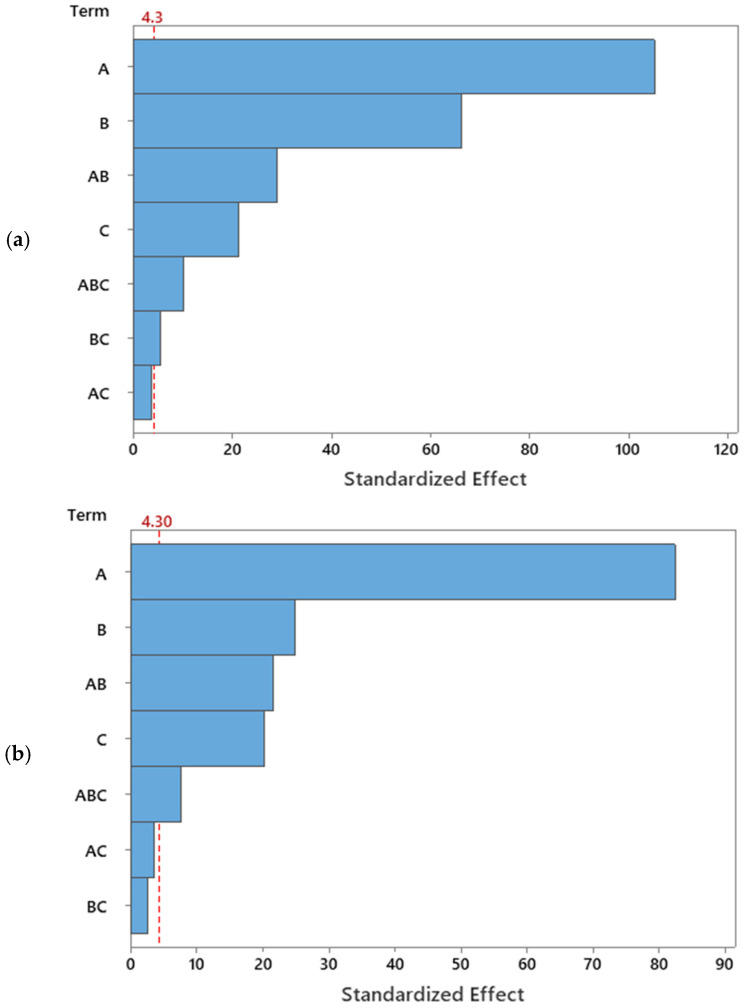
Pareto chart of the standardized effects of (**a**) the membrane flux and (**b**) permeate volume with a = 0.05.

**Figure 4 membranes-14-00199-f004:**
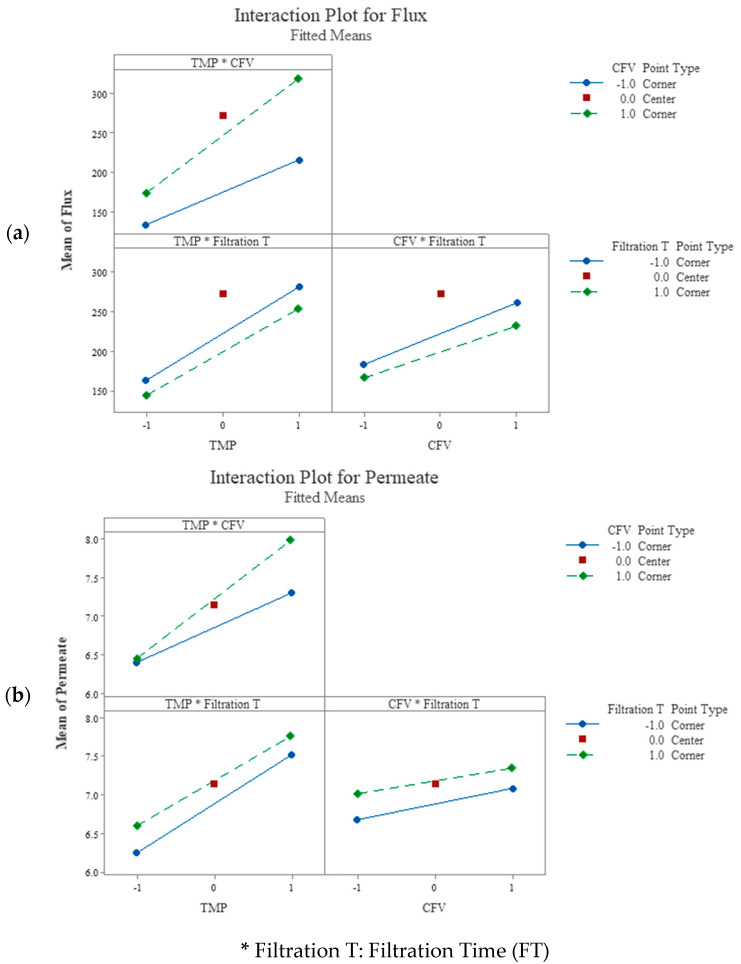
Factorial plots for (**a**) mean membrane flux and (**b**) permeate volume.

**Figure 5 membranes-14-00199-f005:**
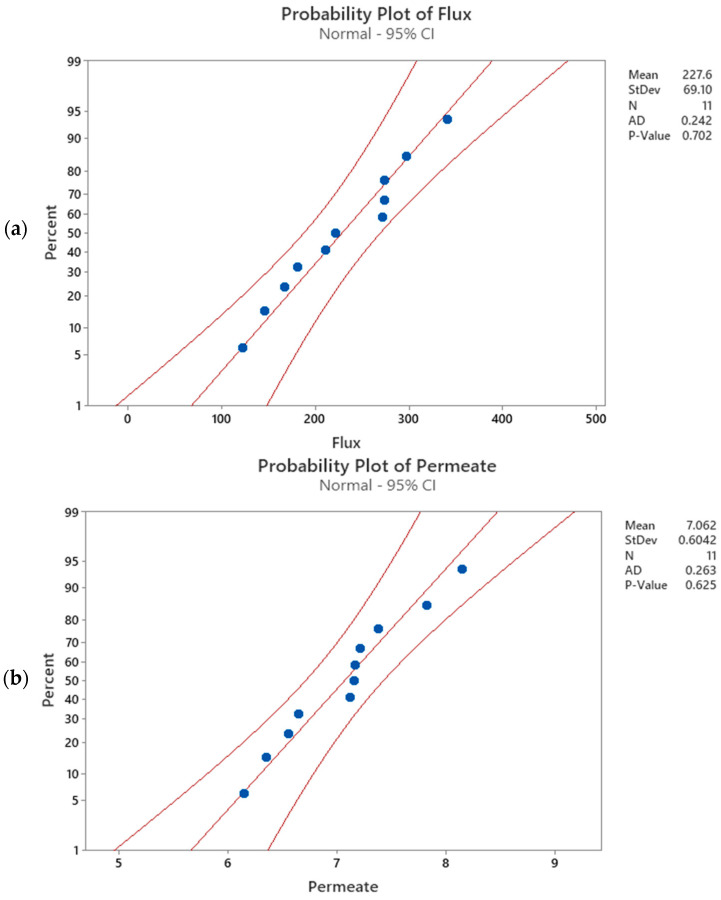
Normal probability plots for (**a**) membrane flux and (**b**) permeate volume.

**Figure 6 membranes-14-00199-f006:**
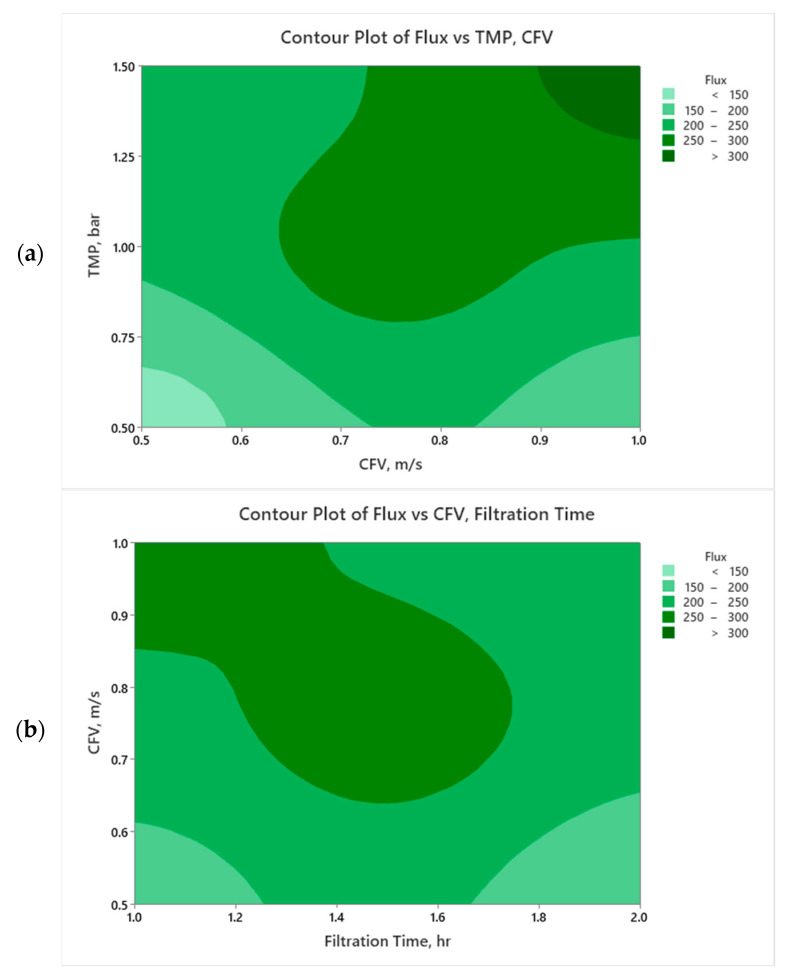
Contour plots for membrane permeate flux as a function of (**a**) TMP and CFV at FT = 1.5 h and (**b**) CFV and FT at TMP = 1 bar.

**Figure 7 membranes-14-00199-f007:**
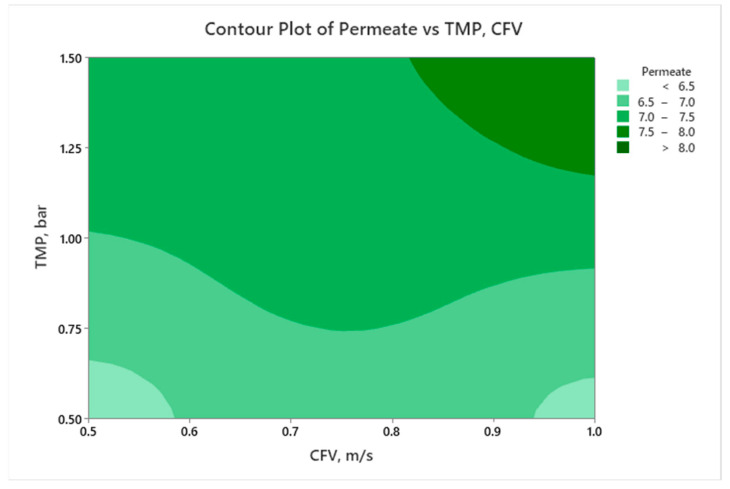
Contour plots for permeate volume as a function of TMP and CFV at FT = 1.5 h.

**Figure 8 membranes-14-00199-f008:**
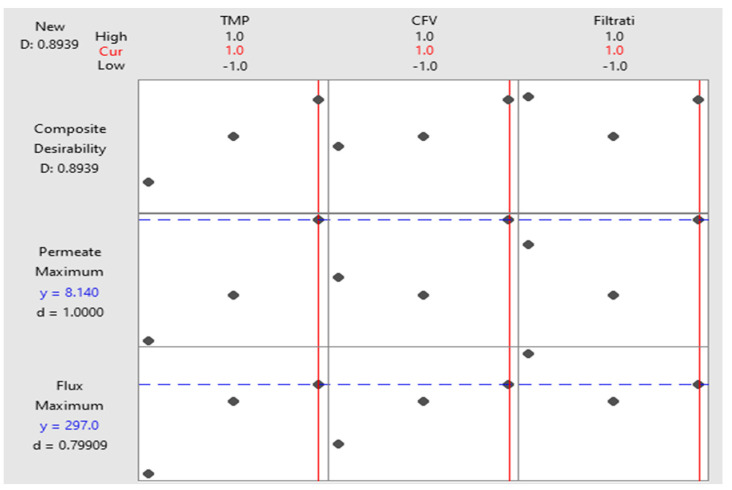
Optimization plot of the operating conditions. The dots present the operating conditions levels (−1, 0, +1) for each factor (TMP, CFV, and FT).

**Figure 9 membranes-14-00199-f009:**
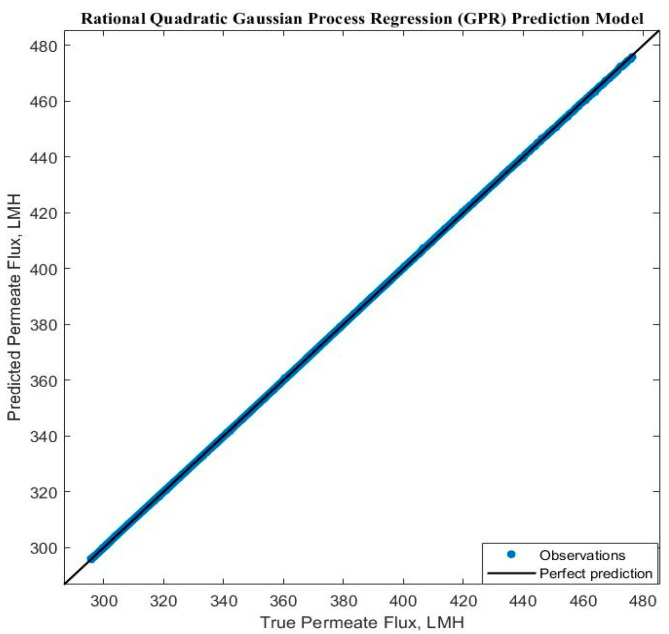
Predicted and real permeate flux correlation response plot.

**Figure 10 membranes-14-00199-f010:**
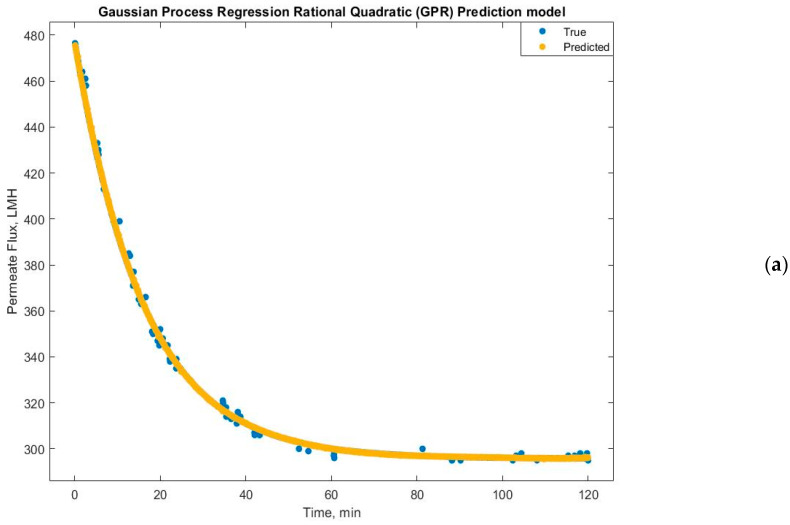
Gaussian process regression quadratic prediction model: (**a**) correlation between true and predicted permeate flux over time, (**b**) recorded and predicted permeate flux at TMP = 1.5 bar, and (**c**) recorded and predicted permeate flux at CFV = 1 m/s during a two-hour experiment.

**Figure 11 membranes-14-00199-f011:**
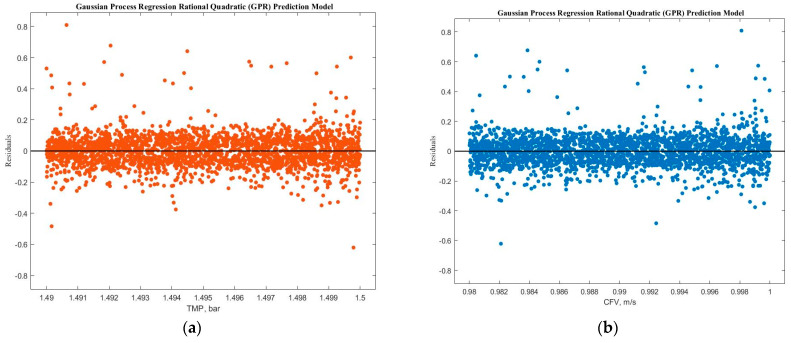
Normalized permeate flux residuals as a function of (**a**) TMP and (**b**) CFV for the experiment.

**Figure 12 membranes-14-00199-f012:**
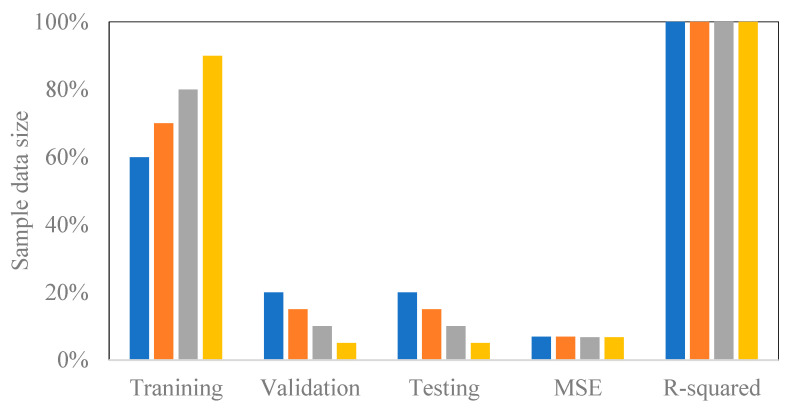
Training performance (R-squared and MSE) as a function of the experimental data percentages training.

**Table 1 membranes-14-00199-t001:** Overview of the operating conditions optimization design methods applied for the membrane processes.

Process	Feed	Operating Conditions	Response	Optimization Methodology	Performance	Refs.
Crossflow filtration unit (LiqTech)	Produced water	TMP, CFV, Temp, and pH	Permeate flux	Taguchi, ANOVA, and ANN	Oil rejection = 98.25%	[2]
Crossflow filtration unit (LiqTech)	Produced water	TMP, CFV, and pulsatile cycle	Permeate flux and Permeate volume	Box–Behnken and ANOVA	99% oil rejection, R^2^ = 99%	[4]
Crossflow filtration unit (LiqTech)	Produced water	TMP, CFV, Temp, and pH	Permeate flux fouling resistance	Taguchi, ANOVA, and ANN	96% oil rejection	[13]
Crossflow filtration unit (LiqTech)	Produced water	TMP, CFV, Temp, and pH	Permeate flux	Taguchi, ANOVA, and ANN	98.93% oil rejection 99% organic carbon removal (TOC) 99.82% turbidity removal	[14]
Crossflow filtration unit (LiqTech)	Produced water	TMP, CFV, Temp, and pH	Permeate flux	Taguchi, ANOVA, and ANN	Oil rejection = 98% Toral organic carbon = 99% Turbidity rejection = 99%	[15]
Crossflow filtration unit	Industrial wastewater	Oil concentration, TMP, and Temp	COD Permeate flux	Box–Behnken and ANOVA	COD removal, R = 0.985 Permeate flux, R = 0.901	[16]
Conventional jar-test apparatus	Vegetable oil refinery	Coagulant concentration, flocculent dosage, and Initial pH	COD removal Residual turbidity	Box–Behnken and ANOVA	COD removal, R^2^ = 92.9% Turbidity removal, R^2^ = 83.6%	[17]
Crossflow filtration unit	Vegetable oil seed wastewater	TMP, CFV, and Temp	Permeate flux Chemical oxygen demand reduction (%)	RSM	COD removal enhanced from 40% to 75%.	[18]
Ozonation-assisted hybrid reactor	Oil and gas oily wastewater	Hydraulic residence time, Aeration, Current density, Intermittent power, and initial pH	COD rejection	One-factor-at-a-time experimentation and RSM	Efficiency was considerably improved, attaining 53.1% COD removal.	[19]
Ultrafiltration crossflow filtration unit	Cutting oil wastewater	Oil content, CFV, TMP, aspect ratio, and (AR) of twisted tape	Permeate flux Energy consumption	Central composite face-centred design	Maximal Steady flux = 201 LMH Minimal specific energy consumption = 1.34 kWh/m^3^	[20]
Laboratory bench plant	Orange press liquor	Contact angle, membrane thickness, pore size distribution, TMP, temperature, and process time	Permeate flux, hesperidin, glucose, fructose, and sucrose rejection	Partial least squares regression (PLSR)	Permeate flux, R^2^ = 96.2% Hesperidin rejection, R^2^ = 95.8% Glucose rejection, R^2^ = 91.7% Fructose rejection, R^2^ = 97.5% Sucrose rejection, R^2^ = 94.3%	[21]
Double crossflow filtration unit (SEPA CF II-GE Osmonics)	Olive mill wastewater	TMP, Temp, and pH	COD removal Total phenolic removal	Full factorial design	COD removal, R^2^ = 83.3% Total phenolic removal, R^2^ = 93.1%	[23]
Refining operating unit	Vegetable oil refineries wastewater	pH, coagulant dose, flocculant dose, and pollutant load	Turbidity COD removal	Full factorial design and ANOVA analysis	Turbidity, R = 0.96 COD removal, R = 0.9	[24]
Conventional crossflow pilot plant	Polyethylene glycol (PEG)	TMP, CFV, and time	Permeate flux	Hermia model and ANN	Permeability, R² = 0.99	[26]
Flocculation and electrocoagulation laboratory unit	Mining oily wastewater	pH, current density, electrolyte concentration, oil concentration, and Electrode gap	COD removal	ANN and Polynomial GA	Polynomial GA, R^2^ = 0.89 ANN, R^2^ = 0.99	[27]
Membrane rotating biological contactor	Synthetic wastewater	Disk Rotational Speed, hydraulic-retention time, and sludge-retention time	Permeability	SVM and ANN	R^2^ = 99%	[30]
Crossflow module (Rayflow 100 Plate and Frame Mode)	Agricultural palm oily wastewater	TMP, pH, and feed oil content	Lignocellulosic permeate flux	ANN and blocking laws	Water recovery = 82% Rejection = 98%	[31]
Rotating biological contactors	Food leftovers wastewater	Disk rotational speed, membrane-to-disk gap, and organic loading rate	Permeability	RSM and ANN	ANN, R^2^ = 0.9982 RSM, R^2^ = 0.9762	[32]
Anoxic–aerobic membrane bioreactor	Domestic wastewater	COD, MLSS, MLVSS, pH, DO, Alkalinity, TN, TP, NO_3_-N, and NH_4_-N	Transmembrane pressure	ANN	R^2^ = 0.850	[33]
SEPA CF forward osmosis cell (Sterlitec)	Distilled water or 0.086 M NaCl solution	The osmotic pressure difference, feed solution (FS) velocity, draw solution (DS) velocity, FS temperature, and DS temperature	Permeate flux	RSM and ANN	ANN, R^2^ = 0.98036 RSM, R^2^ = 0.9408	[34]
Batch experiment apparatus	Drug solution	pH, contact time, temperature adsorbent dosage, and initial triamterene concentration	Naphthalene removal efficiency	ANN-GA and MLR	MSE = 0.0005 R = 0.9856	[35]
Membrane bioreactor (MBR) filtration	Palm oil mill wastewater	Airflow rate, transmembrane pressure, permeate pump, and aeration pump	Permeate flux	RSM and ANN	MSE = 0.00220 R = 0.9906	[36]
Membrane sequencing batch reactor	Produced water	Time, organic loading rate, reaction time, and TDS	COD, TOC, MLSS, Oil in sludge	ANN-MPC	COD removal = 98%. R^2^ = 0.9822	[37]
Membrane bioreactor	Sludge foulants	Morphology, contact angle, surface tension, zeta potential, and separation distance	Interfacial energy	BP ANN and GRNN	Interfacial energy model prediction, R = 100%	[28]
Membrane bioreactor	Water and wastewater	Mixed liquor suspended solid (MLSS), dissolved oxygen (DO), electrical conductivity (EC), and time	Water flux	ANN and ANFIS	ANN, R^2^ = 0.9822 ANFIS, R^2^ = 0.9822	[29]

**Table 2 membranes-14-00199-t002:** Independent factors, their coded symbols, and level values for full factorial design methodology.

Factors	Coded Symbol	Values of Coded Levels		
		Low (−1)	Middle (0)	High (+1)
TMP (bar)	A	0.5	1	1.5
CFV (m/s)	B	0.5	0.75	1
FT (h)	C	1	1.5	2

**Table 3 membranes-14-00199-t003:** Full factorial design factors and their experimental and predicted responses.

		Uncoded Factors			Code Factors			Responses			
Std Run	Run Test Order	TMP	CFV	FT	A: TMP	B: CFV	C: FT	Permeate Flux (J_ni_)		Permeate Volume (Y_nf_)	
								Experimental	Predicted	Experimental	Predicted
		(bar)	(m/s)	(h)				(LMH)	(LMH)	(L)	(L)
11	1	1	0.75	1.5	0	0	0	271	273	7.15	7.14
1	2	0.5	0.5	1	−1	−1	−1	146	146	6.15	6.15
2	3	1.5	0.5	1	1	−1	−1	221	221	7.21	7.22
10	4	1	0.75	1.5	0	0	0	274	273	7.16	7.14
8	5	1.5	1	2	1	1	1	297	297	8.14	8.14
4	6	1.5	1	1	1	1	−1	341	341	7.82	7.82
6	7	1.5	0.5	2	1	−1	1	211	211	7.38	7.39
9	8	1	0.75	1.5	0	0	0	273	273	7.12	7.14
3	9	0.5	1	1	−1	1	−1	181	181	6.35	6.35
7	10	0.5	1	2	−1	1	1	167	167	6.55	6.55
5	11	0.5	0.5	2	−1	−1	1	122	122	6.65	6.65
Optimized design		1.5	1	2	1	1	1	297	297	8.14	8.14

**Table 4 membranes-14-00199-t004:** Full experimental runs feed and permeate characteristics.

		Synthetic Feed		Permeate		Turbidity	
Std Run	Run Test Order	Mean Oil Droplet Size	Oil Content	Oil Content	Rejection	Feed	Permeate
		(μm)	(ppm)	(ppm)	(%)	(NTU)	(NTU)
11	1	6.5	200	9	96	578	3.64
1	2	5.4	194	15	92	596	5.16
2	3	5.3	199	11	94	583	4.89
10	4	5.9	196	11	94	576	4.73
8	5	6.4	199	5	97	559	2.47
4	6	6.3	196	4	98	568	0.83
6	7	6.9	197	12	94	579	4.62
9	8	5.1	198	10	95	586	4.15
3	9	6.8	197	13	93	566	5.09
7	10	5.5	200	14	93	574	5.13
5	11	5.4	197	18	91	592	5.62
Optimized design		6.4	199	5	97	559	2.47

**Table 5 membranes-14-00199-t005:** Chemicals and brine solution for ceramic membrane cleaning.

Chemicals	Usage	Suppliers
Sodium dodecyl sulfate (SDS, 99 wt%)	Feed synthesis	Sigma-Aldrich (St. Louis, MO, USA)
Hydrochloric acid (HCl, SA431-500, 2N)	Oil/solvent extraction	Fisher Chemicals (Waltham, MA, USA)
Horiba S-316 #100690	Oil/solvent extraction solvent	Horiba (Kyoto, Janpan)
Phosphoric acid (H₃PO₄, 85 wt%)	Ceramic cleaning	BDH Chemicals (London, UK)
Sodium hydroxide (NaOH, 95 wt %)	Ceramic cleaning	EMD chemicals (Burlington, MA, USA)

**Table 6 membranes-14-00199-t006:** Feed characteristics.

Oil Parameters	Feed
Oil Content, ppm	197 ± 2
Chemical Oxygen Demand (COD), mg/L	1352 ± 25
Turbidity, NTU	578 ± 11
pH	6.225 ± 0.001
Zeta potential, mV	−32 ± 4.0
Mean droplet size, μm	6.4 ± 0.1
Density, g/cc	0.87844 ± 5 × 10^−5^
Viscosity, cP	5.23 ± 0.01

**Table 7 membranes-14-00199-t007:** Experimental devices.

Equipment	Measured Parameter/Function	Supplier Area
Horiba Oil Content Analyzer (OCMA-350)	Oil content, ppm	Burlington, ON, Canada
Horiba F-55 benchtop meter (Horiba 2003)	pH	Irvine, CA, USA
Hanna turbidity meter (HI 83414, Hanna 2007)	Turbidity, NTU	Leighton Buzzard, UK
Hach DR5000 UV-Vis spectrophotometer	Chemical Oxygen Demand (COD), mg/L	London, ON, Canada
Zetasizer Nano ZS (ZEN3600, Malvern 2009)	Zeta potential, mV	Great Malvern, UK
Zetasizer Nano ZS (ZEN3600, Malvern 2009)	Droplet size, μm	Great Malvern, UK
Brookfield viscometer DV-II +Pro	Viscosity, cP	Middleborough, MA, USA
Anton Paar 5000 DSA 5000 digital	Density, g/cc	Montreal, QC, Canada
RX-5000 refractometer (ATAGO)	Refractive index (RI.)	Toronto, ON, Canada
Waring Commercial MX1000 Series	Blender	Stamford, CT, USA

**Table 8 membranes-14-00199-t008:** Ceramic membrane characteristics.

Membrane		Characteristics
Dimensions, mm		25 ± 1 × 305 ± 1
Number of channels		7
Dimensions	Filtration area, m^2^	0.04186 ± 0.006
	Cross-sectional area, m^2^	0.001172 ± 0.006
	Channel diameter, mm	6.0 ± 0.1
Parameters	Porosity of support	~38%
	Pore size/MWCO	150 kDa
	Maximum working pressure	10 bar
	Best operating pressure	3 bar
	Applied pH scale	0–14
	Max operating temperature	<250 °C
	Thermal shock resistance	ΔT instantaneous < 60 °C (Temperature difference between feed and membrane)
	Materials	Active layer: Titania Support layer: Zirconia

**Table 9 membranes-14-00199-t009:** Membrane permeate flux Analysis of Variance (ANOVA).

Source	DF	Sum of Square	Mean Square	*F*-Value	*p*-Value
Model	8	47,742	5968	2557.6	0.000
Linear	3	37,047	12,349	5292.4	0.000
TMP	1	25,764	25,764	11,042	0.000
CFV	1	10,224	10,224	4381.93	0.001
FT	1	1058	1058	453.43	0.002
2-Way Interactions	3	2088.5	696.2	298.36	0.003
TMP × CFV	1	1984.5	1984.5	850.50	0.001
TMP × FT	1	32.0	32.0	13.71	0.066 *
CFV × FT	1	72.0	72.0	30.86	0.031
3-Way Interactions	1	242.0	242.0	103.71	0.010
TMP × CFV × FT	1	242.0	242.0	103.71	0.010
Curvature	1	8364	8364	3584	0.001
Error	2	4.7	2.3		
Total	10	47,746			

* not significant at a 95% confidence level.

**Table 10 membranes-14-00199-t010:** Membrane permeate volume Analysis of Variance (ANOVA).

Source	DF	Sum of Square	Mean Square	*F*-Value	*p*-Value
Model	8	3.65010	0.45626	1052.91	0.001
Linear	3	3.38744	1.12915	2605.72	0.000
TMP	1	2.94031	2.94031	6785.34	0.001
CFV	1	0.27011	0.27011	623.34	0.002
FT	1	0.17701	0.17701	408.49	0.002
2-Way Interactions	3	0.20994	0.06998	161.49	0.006
TMP × CFV	1	0.20161	0.20161	465.26	0.002
TMP × FT	1	0.00551	0.00551	12.72	0.070 *
CFV × FT	1	0.00281	0.00281	6.49	0.126 *
3-Way Interactions	1	0.02531	0.02531	58.41	0.017
TMP × CFV × FT	1	0.02531	0.02531	58.41	0.017
Curvature	1	0.02741	0.02741	63.25	0.015
Error	2	0.00087	0.00043		
Total	10	3.651			

* not significant at a 95% confidence level.

**Table 11 membranes-14-00199-t011:** Responses’ parameters optimization.

Response	Goal	Lower	Target	Upper	Weight	Importance
Permeate (L)	Maximum	6.15	8.14	8.14	1	3
Flux (LMH)	Maximum	122.00	341.00	341.00	1	3

**Table 12 membranes-14-00199-t012:** Operating conditions and responses optimized fitted values.

Solution	TMP	CFV	FT	Permeate Fit (L)	Flux Fit (LMH)	Composite Desirability
1	+1	+1	−1	7.82	341	0.916
2	+1	+1	+1	8.14	297	0.894

**Table 13 membranes-14-00199-t013:** Performance comparison of Gaussian Process Regression (GPR) models for all the experimental runs.

	Run 1	Run 2	Run 3	Run 4	Run 5	Run 6	Run 7	Run 8	Run 9	Run 10	Run 11
RMSE	0.111581	0.20797	0.14351	0.11066	0.062188	0.10318	0.14963	0.10843	0.14578	0.16317	0.23217
R^2^	0.99	0.99	0.99	0.99	0.99	0.99	0.99	0.99	0.99	0.99	0.99
MSE	0.013412	0.043251	0.020595	0.12246	0.0038674	0.010647	0.02239	0.011757	0.021251	0.063572	0.31526
MAE	0.076448	0.14847	0.097105	0.075797	0.044647	0.070112	0.10718	0.074881	0.099163	0.27709	1.15277

**Table 14 membranes-14-00199-t014:** ANN’s Mean Squared Error and correlation coefficient models.

Runs	Runs Coded Levels	SAMPLES SIZE	MSE	R
1	(0, 0, 0)	1800	7.68518 × 10^−2^	9.99984 × 10^−1^
2	(−1, −1, −1)	1200	8.291179 × 10^−2^	9.99993 × 10^−1^
3	(+1, −1, −1)	1200	8.10541 × 10^−2^	9.99990 × 10^−1^
4	(0, 0, 0)	1800	8.80532 × 10^−2^	9.99981 × 10^−1^
5	(+1, +1, +1)	2400	5.57747 × 10^−2^	9.99979 × 10^−1^
6	(+1, +1, −1)	1200	8.20400 × 10^−2^	9.99966 × 10^−1^
7	(+1, −1, 1)	2400	6.52329 × 10^−2^	9.99989 × 10^−1^
8	(0, 0, 0)	1800	7.74612 × 10^−2^	9.99983 × 10^−1^
9	(−1, +1, −1)	1200	8.30276 × 10^−2^	9.99992 × 10^−1^
10	(−1, +1, +1)	2400	6.17985 × 10^−2^	9.99992 × 10^−1^
11	(−1, −1, +1)	2400	6.67827 × 10^−2^	9.99994 × 10^−1^

## Data Availability

All data are presented in the article.

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
