# Peer review of "A Novel Modeling Optimization Approach for a Seven-Channel Titania Ceramic Membrane in an Oily Wastewater Filtration System Based on Experimentation, Full Factorial Design, and Machine Learning"

_membranes, 2024, doi:10.3390/membranes14090199_

Round 1

Reviewer 1 Report

Comments and Suggestions for Authors

#1

Figure 13. seems to be all the same for readers. Is it necessary for the authors to show all of the eleven epxerimental runs? 

#2

There's no actual images of the "membrane" itself. Defintely, the authors focused on the novel modeling optimization approach. Although, the results are supported by experimetental results too, there's any kind of SEM images , before/after filtration optical images and so on. 

#3 

Is this research can be applied to any kind of UF membranes or still further researches are needed to enlarge application areas beyond oily wastewater?

#4

Why a seven-channel titania ceramic membrane is selected for this research, the authors needs to address in detail.

Reviewer 2 Report

Comments and Suggestions for Authors

Produced water is created in huge quantities that destroy the environment, which is why the issue of its purification is current and important. Complex and multi-stage purification systems are too expensive to be implemented. Membrane techniques offer hope for developing much cheaper technologies. Unfortunately, after a short period of operation, the membranes become contaminated and the development of an effective method of reducing fouling determines the possibility of industrial implementation.

Describing the mechanisms of fouling is important, but the assumption that knowing them will eliminate them is usually not confirmed in practice. Such works make a significant contribution when their effect is to indicate how to limit fouling. An example is Table 1 - these are good studies for the scientific development of PhD students, but what does it mean for the industry? In the presented work, the authors should describe the development of membrane technology and not mainly focus on improving numerical methods. In its current form the work is limited to readers involved in modeling.

For a given type of wastewaters, there will be different fouling. Hence, the basic conclusion from the work is that the calculation method presented by the authors will allow for a good analysis of the results of pilot studies conducted at a given extraction site. However, the description of the results obtained requires explanation in many places, for example:

1)      Table 3 – why were such low flow velocity values used? For ceramic membranes, we usually use over 3 m/s.

2)      Horiba apparatus tests – samples contained SDS, how was the impact on the oil content measurement results limited? Was there a formation of a thick emulsion?

3)      MWCO – better is kDa

4)      It is not clear – these 11 experiments were done each time for a new membrane?

5)      Cleaning agents were indicated – it should be described when the membranes were washed, what the effect was, whether irreversible fouling occurred, etc.

6)      Fig.6 – should be better discussed, especially clarifying issues that contradict the known state of knowledge:

- 6a – CVF increase at TMP=const not only constant increase in flux, and for larger CVF even its decrease?

- 6b – with time fouling increases and flux stabilizes - meanwhile the authors suggest that after e.g. 1 h the membranes self-clean??? Which is contradicted by fig. 6c

Similar comments to Fig.7

7) Fig.10 b and 10c – the changes shown are not a function of TMP and CFV but result from the elapsed time of the process – correct the signature

Round 2

Reviewer 2 Report

Comments and Suggestions for Authors

The explanations provided only partially explained the comments made in the review. We are examining various processes, also on a pilot scale. However, over the course of many years of research, we have never observed the phenomena suggested by the authors, i.e. that the flux deteriorates due to fouling, and then suddenly improves significantly on its own.

Therefore, the work requires explanation.

Other notes:

-) total permeate volume of 8.14 L. - if we cut the membrane into 0.5 m, it would be e.g. 10 L. That's why we rather use the concentration or recovery rate in the description.

-) turbidity (TNU), improve on NTU

-) membrane's critical pressure  - we rather use the concept of critical flux

-) Fig.6a

the advantages of increased crossflow velocity are limited (Figure 6 (a)). – OK for critical flux

but the origin of the CFV influence alternations shown in the figures is unclear.

e.g. for TMP=0.9 bar at CFV >0.65 m/s the flux increases from 250 to 300 LMH, which is OK, but why above 0.88 m/s does it decrease again to 250 LMH?

Fig.6b – over 1.15 bar is OK,

However, for e.g.1bar the flux is smaller, then increases and after some time it is smaller again. This cannot be explained by the increase in fouling.

The results obtained depend simultaneously on the three parameters tested. It is not clear what values of the third parameter were used to make the figures showing the effects of the two parameters.

Round 3

Reviewer 2 Report

Comments and Suggestions for Authors

The authors presented various modeling approaches and this part of the work is correct.
However, I am convinced that the amount of experience, i.e. data for modeling, is too small to demonstrate whether the presented fluctuations and wobbles in the results can actually occur.
Therefore, I would suggest removing these fragments of the work.

Author Response

We moved some of the figures to the supplementary material section as the p obtained in terms of the interactions was less than 0.05.

The changes in the flux at constant TMP are explained in the manuscript (468-513) as it was done in multiple other studies (see 2 examples in the attached file). We attached 2 examples of similar contours. The literature has a dozen similar contour analyses but shown as 3D. These contours are surface contours obtained using ANOVA. The raw data about the flux as a function of time is presented in Figure 10 a for 1 run.
